# Intelligent Detection of Hazardous Goods Vehicles and Determination of Risk Grade Based on Deep Learning

**DOI:** 10.3390/s22197123

**Published:** 2022-09-20

**Authors:** Qing An, Shisong Wu, Ruizhe Shi, Haojun Wang, Jun Yu, Zhifeng Li

**Affiliations:** 1School of Artificial Intelligence, Wuchang University of Technology, Wuhan 430223, China; 2China Railway Wuhan Survey and Design Institute Co., Ltd., Building E5, Optics Valley Software Park, No. 1, Guanshan Avenue, Donghu High-Tech Zone, Wuhan 430050, China; 3School of Safety Science and Emergency Management, Wuhan University of Technology, Wuhan 430070, China; 4USTC iFLYTEK Co., Ltd., Hefei 230088, China

**Keywords:** deep learning, EfficientDet, vehicle detection, learning rate

## Abstract

Currently, deep learning has been widely applied in the field of object detection, and some relevant scholars have applied it to vehicle detection. In this paper, the deep learning EfficientDet model is analyzed, and the advantages of the model in the detection of hazardous good vehicles are determined. The adaptive training model is built based on the optimization of the training process, and the training model is used to detect hazardous goods vehicles. The detection results are compared with Cascade R-CNN and CenterNet, and the results show that the proposed method is superior to the other two methods in two aspects of computational complexity and detection accuracy. Simultaneously, the proposed method is suitable for the detection of hazardous goods vehicles in different scenarios. We make statistics on the number of detected hazardous goods vehicles at different times and places. The risk grade of different locations is determined according to the statistical results. Finally, the case study shows that the proposed method can be used to detect hazardous goods vehicles and determine the risk level of different places.

## 1. Introduction

The safe production of road transportation for hazardous goods is related to the safety of life and property of the country and people, as well as national economic development and social harmony and stability. Hazardous goods are in a hazardous process, from coming out of the warehouse to arriving at the destination through vehicle transportation. Therefore, it is particularly important to supervise the transportation process of hazardous goods vehicles. The supervision of the transportation process of hazardous goods vehicles involves supervision of vehicle travel dynamics, the frequency of vehicles passing through a certain place, and accidents. The dynamic supervision of hazardous goods vehicle travel is realized mainly based on the GPS positioning of the hazardous goods vehicle. The specific frequency of passing through a certain place and accident conditions of the hazardous goods vehicles can be achieved by the number of times and continuous detection time of the hazardous goods vehicle detected by the camera. Due to the influence of environmental factors such as lighting conditions, partial occlusion, and messy background, the accuracy of hazardous goods vehicle detection will be greatly affected. In order to improve the reliability and accuracy of hazardous goods vehicle detection, foreign scholars have carried out a great deal of research on it.

Vehicle detection methods mainly include image-based detection methods and deep learning-based detection methods. The image-based detection method mainly detects vehicle targets through vehicle image features and directional gradient histogram features. For example, Arthi R et al. [1] used the feature transformation of the image for vehicle classification and detection. Matos F et al. [2] achieved vehicle detection by analyzing the edge features of vehicle images and combining principal component analysis. Although the vehicle detection method based on image features has low computational complexity and can detect vehicles quickly, it is difficult to detect it in the area where the vehicle has partial occlusion or illumination change. In view of this, Iqbal U et al. [3] enhanced vehicle image features by fusing Sobel and SIFT features to realize vehicle detection. M.T. Pei et al. [4] used Sobel edge detection to detect vehicles in parking spaces as well as realized the detection scores of different types of vehicles. S. Ghaffarian [5] used a classifier based on fuzzy c-means clustering and super parameter optimization to detect vehicles and completed the location of vehicles in a parking space based on the vehicle detection results. Because the image of the vehicle itself has unique texture features, vehicle detection can be realized according to the vehicle texture features [6]. At present, the main disadvantage of vehicle detection methods based on vehicle image texture features and vehicle edge features is that they are greatly affected by illumination and vehicle integrity. With the continuous development of deep learning [7,8,9,10], more and more scholars have begun to study vehicle detection based on the deep learning method. X.J. Shen et al. [11] used a convolutional neural network to train vehicle images and trained models for vehicle detection. X. Xiang et al. [12] proposed a vehicle detection method based on the Haar–Adaboosting algorithm and convolutional neural network. Tang T et al. [13] proposed a super region candidate network, which can detect small vehicles photographed by distant cameras. In the process of vehicle detection, the vehicle is easily obscured by other objects. In order to solve the problem that the vehicle can be detected in the case of occlusion, Wang X et al. [14] introduced countermeasure learning into the process of RCNN target detection, which improves the accuracy of vehicle detection. The advantage of the RCNN algorithm is high detection accuracy, but its detection speed is slow, so it is difficult to implement real-time detection of vehicles. In order to improve the detection efficiency, Lu J et al. [15] introduced Yolo series algorithms to realize vehicle detection. The detection network is mainly based on an SSD network, which can detect vehicles quickly, but its accuracy is not particularly high. In order to improve the detection accuracy, Cao G et al. [16] integrated cascade modules and element modules to improve the SSD network and realize high-precision vehicle detection. However, due to the integration of more modules, the detection speed decreases. At present, deep learning [17,18,19] has received extensive attention in the field of vehicle detection. In the process of vehicle detection, a two-time-scale discrete-time system with multiple agents was used to optimize multi-vehicle detection [20]. Therefore, this paper will also use the deep learning method to implement vehicle detection. Specifically, we optimize the training stage of the deep learning EfficientDet model and build a phased training model to realize fast and accurate vehicle detection.

## 2. Construction of Vehicle Detection Model

Firstly, the effective deep learning model is used to train the hazardous goods vehicles, and the trained model is used to detect the hazardous goods vehicles. The target recognition network based on deep learning mainly includes cascade R-CNN [21], SpineNet [22] and CenterNet [23,24]. In order to verify the performance of different deep learning networks, Table 1 shows the detection performance of different recognition network models. COCO mAP [^1] in Table 1 is the average accuracy that is used to measure detection performance. The larger the value of mAP, the better the detection performance [25].

From Table 1, it is obvious that the detection performance of the deep learning network model of the EfficientDet series is the best, followed by the cascade R-CNN_ResNet deep learning network model. The best detection performance of the EfficientDet series is the EfficientDet-D7x network. The recognition accuracy of this network reaches 55.1, followed by EfficientDet-D3. The disadvantage of the EfficientDet-D7x network is that the complexity is too high to be used as the demand for real-time vehicle detection. In view of this, this paper selects EfficientDet-D3 as the vehicle detection network.

In order to build a hazardous goods vehicle detection model that is optimized based on DfficientDet-D3, the BiFPN depth named DbiFPN is linearly increased, and the BiFPN named width WbiFPN is exponentially increased. The depth and width of BiFPN are obtained, as shown in Equation (1).
(1)$$\left\{\begin{array}{c} D_{BiFPN}=3+\emptyset \\ W_{BiFPN}=64*\left(1.35\emptyset \right) \end{array}\right.$$

For the EfficientDet-D3 regression prediction network, the regression prediction box width is fixed to be the same as BiFPN ($W_{pred}=W_{BiFPN}$), and Equation (2) is used to linearly increase its depth.
(2)$$D_{box}=D_{class}=3+\left[\emptyset /3\right]$$

In order to increase the image detection accuracy, the input image resolution needs to be increased. Considering the BiFPN function level in the EfficientDet-D3 deep learning network, Equation (3) is used to linearly improve the resolution.
(3)$$R_{input}=512+\left[\emptyset *128\right]$$

Since ∅=3, the construction of the EfficientDet-D3 deep learning network model was completed. Figure 1 shows the EfficientDet-d3 network structure, in which BiFPN, as a feature network, obtains features and reuses top-down and bottom-up bidirectional feature fusion. These fused features are sent to the classification and box regression network to generate the target category and predicted bonding box. We employed ImageNet-pre-trained EfficientNets as the backbone network. The BiFPN serves as the feature network, which takes level 3–7 features $\left\{P_{3},P_{4},P_{5},P_{6},P_{7}\right\}$ from the backbone network and repeatedly applies top-down and bottom-up bidirectional feature fusion. The classification and box regression network weights are shared among the features of all levels.

Based on the EfficientDet-D3 deep learning network model, this paper establishes a hazardous goods vehicle detection model, as shown in Figure 2. In Figure 2, the EfficientDet backbone and BiFBN layer were first used to construct the deep learning network. Then, $D_{BiFPN}$ and $W_{BiFPN}$ were used to conduct the classification of vehicles. Simultaneously, the vehicle detection model is constructed according to Equation (4). Finally, we used the vehicle detection model to train the hazardous goods vehicles and detect them.
(4)$$\begin{array}{c} L_{conf}\left(x,c\right)=-\sum_{i\in Pos}^{N}x_{ij}^{p}\log\left({\hat{c}}_{i}^{p}\right)-\sum_{i\in Neg}\log\left({\hat{c}}_{i}^{0}\right)\\ {\hat{c}}_{i}^{p}=\frac{\exp(c_{i}^{p})}{\sum_{p}\exp(c_{i}^{p})} \end{array}$$
where $x_{ij}^{p}=\left\{1,0\right\}$ is the matching degree of the *i*-th a priori detection frame to the *j*-th real detection frame in category *p*, and $c_{i}^{p}$ is the predicted value of category confidence of the *i*-th a priori detection box.

## 3. Experiment Analysis

### 3.1. Experiment Settings

The vehicle image data sets under different scenes in different time periods are collected, as shown in Figure 3. The data set is divided into the training data set, verification data set, and test data set. Among them, 2387 images are training data sets, 211 images are verification data sets, and 146 images are test data sets. A total of 2744 are annotated with labeling tool. The annotation file generated by each image is converted into Tensorflow unified TFRecord data format through a Python script file. The server configuration used for the experiment is an NVIDIA 3070 8G graphics card with 32G of RAM and an Intel^®^ Core™ i7-10,700F CPU (Santa Clara, CA, USA). The operating system is Ubuntu 22.04 and CUDA 11.4 for the parallel computing framework.

For the vehicle detection, there are four possible detection outcomes, as listed in Table 2.

In Table 2, TP indicates the number of correctly detected vehicles; FP indicates the number of incorrectly detected vehicles; FN indicates the number of vehicles that are missed; TN indicates the number of correctly detected non-vehicles. According to the four possible detection results, we use Precision, Recall and F1-score to evaluate the performance of the model, defined as
(5)$$\mathrm{Precision}=\frac{\left|\mathrm{TP}\right|}{\left|\mathrm{TP}\right|+\left|\mathrm{FP}\right|}$$
(6)$$\mathrm{Recall}=\frac{\left|\mathrm{TP}\right|}{\left|\mathrm{TP}\right|+\left|\mathrm{FN}\right|}$$
(7)$$F1-\mathrm{score}=2\times \frac{\mathrm{precision}\times \mathrm{recall}}{\mathrm{precision}+\mathrm{recall}}$$

### 3.2. Training of the Model

According to the characteristics of the data set, the parameters in the configuration file corresponding to the model are adjusted before training, including the number of classes, batch size, initial learning rate, and related data reading path. The hazardous goods vehicle detection model is not affected by the vehicle model. At the beginning of the iteration, some data are selected, and the predicted value is obtained through the deep learning algorithm [26,27,28,29].

According to the data set size and computer configuration, we first selected the batch size of the initial training and then adjusted it according to the change of loss function value and detection effect. The standard gradient descent algorithm was used to train the hazardous goods vehicle detection model. In the training stage of the model, 2387 images were used to train the model, and 211 images were used to verify the effect of model training. We used the TensorFlow_Slim module to construct the model, which provides a simple but powerful training model. In the process of model training, the return value of the loss function is the value of the objective function generated in each iteration. In addition, the sum of vehicle positioning loss and confidence loss of the detection model is an index to measure the performance of the prediction model. In order to debug and optimize the model training process, TensorFlow provides a visual tool TensorBoard, which monitors and displays the training process by reading the recorded data file. The larger the batch size, the higher the memory utilization and the faster the data processing speed. In training, when using a single NVIDIA GeForce RTX 2080Ti GPU (Santa Clara, CA, USA), this article sets the batch size to 16. Learning rate is one of the most important parameters affecting the performance of the model. Too large of learning rate will lead to unstable detection, and too small of learning rate will lead to over fitting and slow convergence. In view of this, this paper uses AdamW algorithm [30] to optimize the learning rate. In the training stage, if the loss does not decrease for three consecutive cycles, the learning rate will decrease. In this paper, the changes of learning rate parameters in the training stage of hazardous goods vehicle detection model are adjusted according to the variations of total loss value, as shown in Figure 4.

According to Figure 4, the first stage is the iteration stage from 0 to 41,870, and the initial learning rate was set to 1e-3. Simultaneously, it can be seen from Figure 5 that the total loss value decreased rapidly to 0.26, which means that the learning rate needed to be adjusted. The second stage iterated from 41,871 to 54,980, and the learning rate was set to 1e-4 in order to determine whether the learning rate still needed to be optimized. It can be seen from Figure 5 that the total loss value was reduced to 0.225 and tends to be smooth nearby. Finally, from 59,431 iterations to the end, the learning rate was reduced to 1e-6, and the total loss value remained unchanged, which was 0.225, and tended to be smooth. After all the iterations, the whole model training process was completed, the total training convergence time was about 37 h, the training score was greater than 99.5%, the total learning rate parameters were adjusted three times, and finally, the construction of the deep learning vehicle detection model was completed.

### 3.3. Ablation Experiments

In order to verify that the proposed method have an effect on the accuracy and speed of the EfficientDet-D3 model, the ablation experiment is designed to verify its effectiveness. We use the improved EfficientDet-D3 and original EfficientDet-D3 to conduct the detection of hazardous goods vehicles. In the training phase, 211 images are used to verify the effect of these two models’ training, and the training time and accuracy of these two methods are obtained, as shown in Table 3.

Table 3 shows that the improved EfficientDet-D3 needs spend 4.2 h to finish the training of 2387 images. However, the original EfficientDet-D3 takes 6.3 h to complete the training of 2387 images. It is clearly visible that the training time of the improved EfficientDet-D3 model is much less than the original EfficientDet-D3 model. It illustrates that the proposed method improves the training efficiency and shortens the training time. Additionally, the training accuracy of the improved EfficientDet-D3 is almost the same as the original EfficientDet-D3. It shows that the optimization method in this paper does not affect the model training accuracy.

We use the original and improved EfficientDet-D3 to conduct the detection of 146 images. According to Equations (5)–(7), we obtain the Precision, Recall, and F1-scores of these two methods, as shown in Table 4.

Table 4 shows that the precision of the original EfficientDet-D3 is the same for Recall and F1-score. The reason for this phenomenon is that the sum of TP and FP is equal to the sum of TP and FN. From Table 4, it is clearly visible that the precision, Recall, and F1-score of improved EfficientDet-D3 are greater than original EfficientDet-D3. It illustrates that the performance of the improved EfficientDet-D3 is better than the original EfficientDet-D3 for the detection of hazardous goods vehicles.

### 3.4. Performance Analysis

(1)Comparison of different detection methods for hazardous goods vehicles

In order to verify the performance of the proposed method, we use 146 test data sets images to conduct the detection of hazardous goods vehicles and compare the detection results with cascade R-CNN, CenterNet, and EfficientDet-D7x methods. The 146 test data sets include long-distance and short-range hazardous goods vehicles. Additionally, the 146 test data sets are composed of 102 vehicle images and 44 non-vehicle images. Table 5 shows the number of parameters to be used in the detection of hazardous goods vehicles by these four methods.

From Table 5, it is clearly visible that the parameter of the proposed method is much lower than Cascade R-CNN, CenterNet, and EfficientDet-D7x methods. The proposed method implements the detection of hazardous goods vehicles with a minimum number of parameters. The parameter of the Cascade R-CNN is much higher than the other three methods. The Cascade R-CNN uses the maximum number of parameters for detecting hazardous goods vehicles. The parameter of CenterNet is almost half of Cascade R-CNN. However, it is still higher than EfficientDet-D7x and proposed methods. It illustrates that the calculation complexity of the cascade R-CNN method is the highest, followed by the CenterNet model. Simultaneously, the calculation complexity of the proposed method is much lower than the other three methods.

We used the four methods to conduct the detection of hazardous goods vehicles. According to the concept of TP, FP, TN, and FN, the detection results of the 146 test data sets images of different methods were obtained, as shown in Table 6.

Table 6 shows that the TP values of Cascade R-CNN and CenterNet methods are lower than EfficientDet-D7x and proposed methods. The main reason for the detection results is that the detection effect of Cascade R-CNN and CenterNet methods are easily affected by long-distance hazardous goods vehicles. The FP values of the Cascade R-CNN, CenterNet, and EfficientDet-D7x methods are greater than the proposed method. The possible reason for this phenomenon is that these three methods are easily influenced by the non-vehicle images and identify the vehicle-like objects as vehicles.

According to the TP, TN, FP, and FN, the tri-partite measures are calculated, as shown in Table 7.

The precision of the CenterNet is lower than the other three methods. However, its Recall is higher than Cascade R-CNN. The reason is that the precision is related to TP and FP, and the Recall is related to TP and FN. The FN of CenterNet is lower than Cascade R-CNN. The precision, Recall, and F1-score of the EfficientDet-D7x and proposed method are higher than Cascade R-CNN and CenterNet methods. Although, the Recall of the proposed method is slightly lower than EfficientDet-D7x. The precision of the proposed method is higher than EfficientDet-D7x. Simultaneously, the F1-score of the proposed method is slightly higher than EfficientDet-D7x. The precision and F1-score of the proposed method are both considerably higher than the other methods. Therefore, the experimental results show that the proposed method can successfully improve the vehicle identification accuracy compared with Cascade R-CNN, CenterNet, and EfficientDet-D7x by reducing false identifications.

These four methods are used to detect hazardous goods vehicles, as shown in Figure 6.

As can be seen from Figure 6, the detection scores of these four methods for short-range vehicles are greater than 80%. It shows that these four methods are suitable for vehicle detection in this case. Among them, the detection scores of the two hazardous goods vehicles by this method are 98% and 99%, which is higher than Cascade R-CNN and CenterNet methods. Although, it is slightly lower thanEfficientDet-D7x method. The computational cost and memory requirements of the proposed method are much lower than the EfficientDet-D7x method.

In order to evaluate the detection efficiency of the proposed method and the other three methods in detecting hazardous goods vehicles, the time cost of detecting hazardous goods vehicles is obtained, as shown in Figure 7. As can be seen from Figure 7, the detection time of the Cascade R-CNN is the longest, which takes about 160 milliseconds. The detection time of the EfficientDet-D7x is similar to that of the method based on CenterNet, and the two methods only take about 40 milliseconds, which is completely lower than that of the method based on Cascade R-CNN. The detection time of the proposed method is much lower than the Cascade R-CNN and slightly lower than the CenterNet and EfficientDet-D7x methods. Therefore, from the detection time cost, it can be seen that the proposed method is better than the other three methods.

(2)Comparison detection of hazardous goods vehicles in different scenarios

In order to evaluate that the deep learning model constructed in this paper is applicable to the detection of hazardous goods vehicles in different scenarios, the hazardous goods vehicles in different scenarios are detected, as shown in Figure 8. The higher the detection score of hazardous goods vehicles, the better the detection effect.

It can be clearly seen from Figure 8 that the vehicle detection scores under six different backgrounds are greater than 90%, especially when there is interference from other objects around the vehicle body. The vehicle detection score of hazardous goods is still greater than 90%, while the background interference object detection score is less than 50%, as shown in Figure 8d,f. This shows that this method has a good effect on the detection of hazardous goods vehicles and is suitable for the detection of hazardous goods vehicles under different backgrounds.

## 4. Case Study

There are four hazardous goods warehouses in the Hanyang district of Wuhan, as shown in Figure 9. According to the route of hazardous goods vehicles marked in the figure, it is clear that all vehicles must pass the yellow route in the upper left corner, as shown in the location of camera one. We used the trained deep learning models to conduct the detection of hazardous goods vehicle on cameras in ten locations.

We set the detection period to one week, and the number of hazardous goods vehicles were counted every day, as shown in Figure 10.

From Figure 10, it is visible that the number of hazardous goods vehicles in location one is much greater than in location two. The reason for this result is that all the hazardous goods vehicles from hazardous goods warehouses pass through location one. The number of hazardous goods vehicles in location two is almost the same as in locations three and four. It illustrates that the number of distributed vehicles on the three roads is similar. Location 10 has the least number of hazardous goods vehicles. The reason for this phenomenon is that the hazardous goods vehicles only from one hazardous goods warehouse pass location 10. There are far fewer hazardous goods vehicles on Saturdays than on other days. The main reason is that Saturday is a rest day and only a few hazardous goods vehicles are in and out.

From Figure 11, it is visible that hazardous goods vehicles pass through locations 1 to 10. It illustrates that positions 1 through 10 are all possible hazardous locations affected by hazardous goods. The number of hazardous goods vehicles in location 1 is much greater than in other locations. It illustrates that the location one has the highest risk of being affected by hazardous goods. The risk of position 2 is like that in positions 3 and 4, and they are higher than in positions 5 to 10. The levels of risk descend from location 5 to location 10. According to Figure 11, it is clearly visible that the risk level of location 10 is lower than other locations.

According to the statistical result, we can use the total number of hazardous goods vehicles to evaluate the risk level of each location. The risk for different locations is divided into six levels, and different risk levels are labeled by circles of different sizes and colors, as shown in Figure 12.

Figure 12 depicts the different risk levels of different locations. The size of the circle on location one is the largest, and the color is red. It is visible that location one is the highest risk level affected by hazardous goods. The reason is that all the hazardous goods vehicles need to pass through position 1. Locations two, three, and four have the same risk level of 2. This is followed by position three, with a risk level of 3 and a circle size smaller than position one, two, three, and four, colored yellow. The risk level of position six is 4. The risk level of position seven is the same as position eight and nine. The risk level of position ten is 6, the lowest risk. Therefore, risk levels at different locations can be determined according to the number of hazardous goods vehicles. Simultaneously, the risk level of the road section passed by hazardous goods vehicles can be clearly seen in the map.

## 5. Conclusions

In this paper, the hazardous goods vehicle detection method based on deep learning is proposed, and a hazardous goods vehicle detection model based on the Efficientdet-d3 model is established. In the training stage of the Efficientdet-d3 model, to improve the training efficiency of the detection model, the setting of phased training and learning parameters is given according to the change of total loss value. The learning mechanism of adaptive model training is established.

Comparing the detection model in this paper with the methods based on cascade R-CNN and CenterNet, the method in this paper uses the least parameters and has the lowest computational complexity. The detection time of hazardous goods vehicles in this method is equivalent to that of the CenterNet method, which is completely lower than that of the cascade R-CNN method. At the same time, the detection accuracy of the three methods is basically the same. Finally, from the three aspects of computational complexity, time consumption and detection accuracy, it is determined that this method is better than the other two methods.

The detection model is used to judge hazardous goods vehicles in different scenes. The results show that this method can accurately detect hazardous goods vehicles in different scenes, and the detection accuracy is higher than 90%. The deep learning model constructed in this paper is applied to the detection of hazardous goods vehicles in four sections of Wuhan Petrogoods Company. The experimental results show that the accuracy of this method is higher than 90%. It shows that this method can be used to detect hazardous goods vehicles in different sections. This paper analyzes the detection of hazardous goods vehicles in the surrounding sections of four hazardous goods warehouses in the Wuhan District of Wuhan and obtains the number of hazardous goods vehicles passing through each section according to the detection results in one week. The risk level of different sections can be obtained according to the passing times of hazardous goods vehicles in different sections, and then the risk level of each section around the hazardous goods warehouse can be clearly seen on the map.

## Figures and Tables

**Figure 1 sensors-22-07123-f001:**
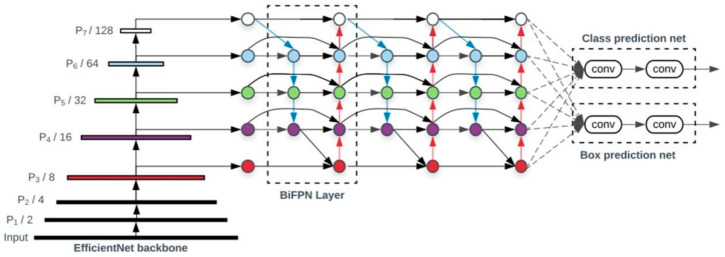
EfficientDet-D3 deep learning network.

**Figure 2 sensors-22-07123-f002:**
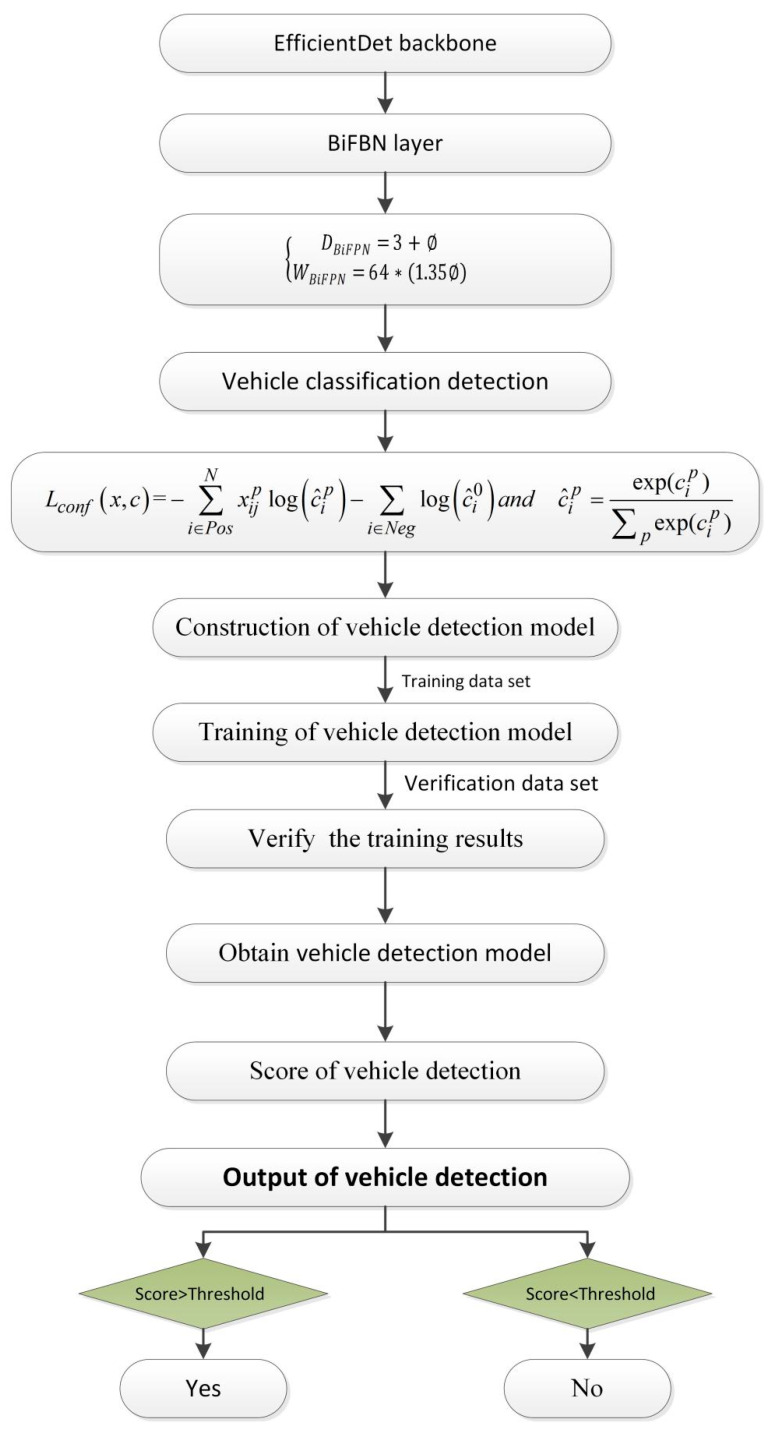
Vehicle detection diagram.

**Figure 3 sensors-22-07123-f003:**
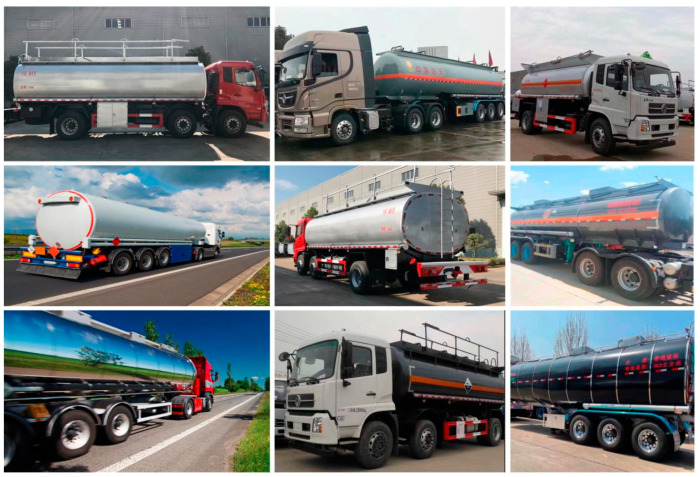
Vehicle data set.

**Figure 4 sensors-22-07123-f004:**
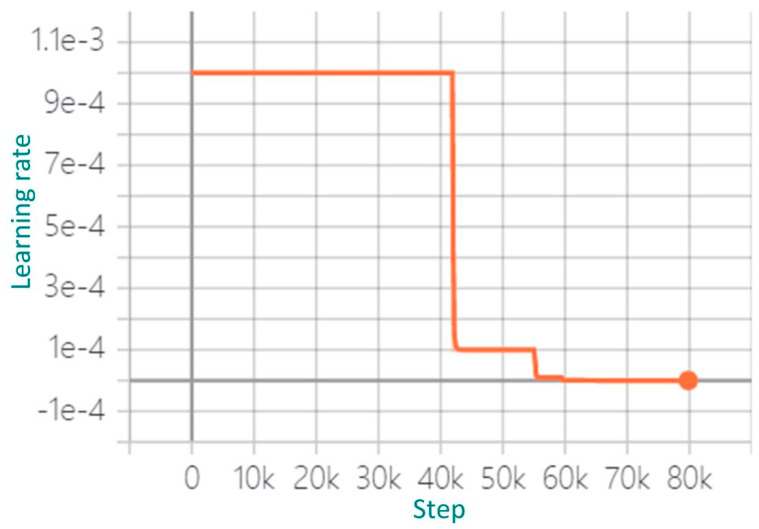
Variation of learning rate parameters.

**Figure 5 sensors-22-07123-f005:**
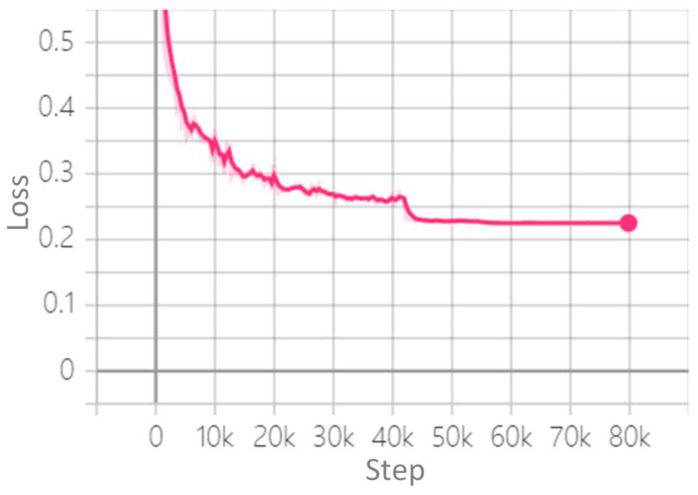
Variation trend of total loss value.

**Figure 6 sensors-22-07123-f006:**
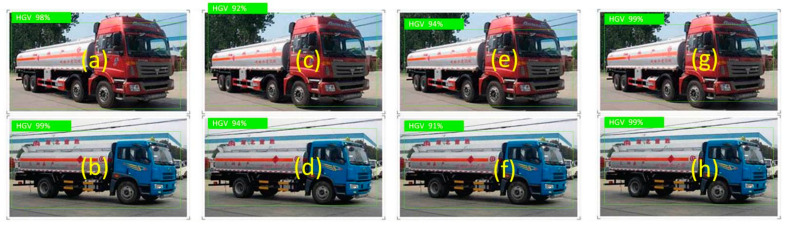
Detection effect of different methods on hazardous goods vehicles. (**a**,**b**) Proposed method. (**c**,**d**) Cascade R-CNN method. (**e**,**f**) CenterNet method. (**g**,**h**) EfficientDet-D7x method.

**Figure 7 sensors-22-07123-f007:**
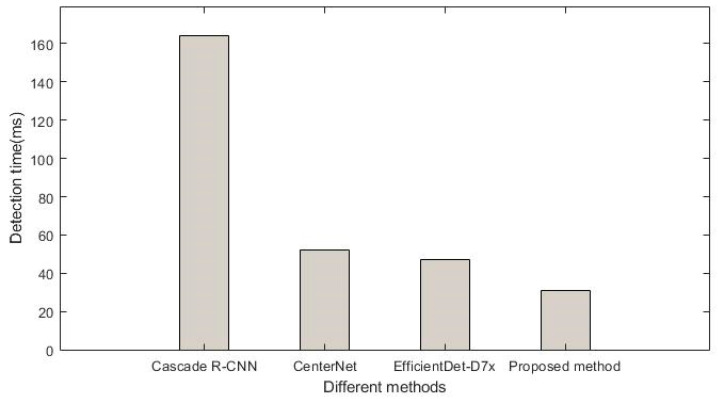
Detection time of four different methods.

**Figure 8 sensors-22-07123-f008:**
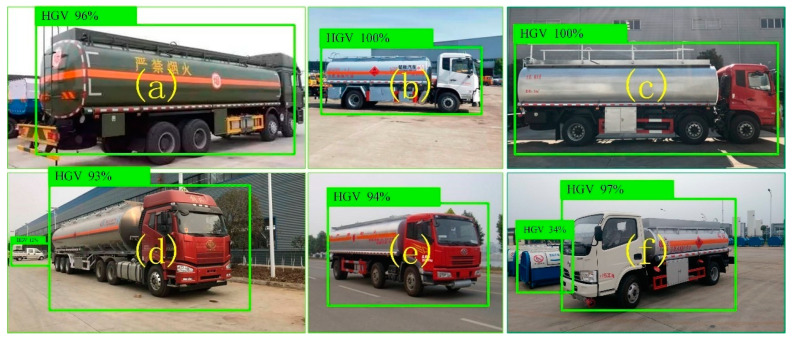
(**a**–**f**) Deep learning vehicle detection model for hazardous goods vehicle detection in different scenarios.

**Figure 9 sensors-22-07123-f009:**
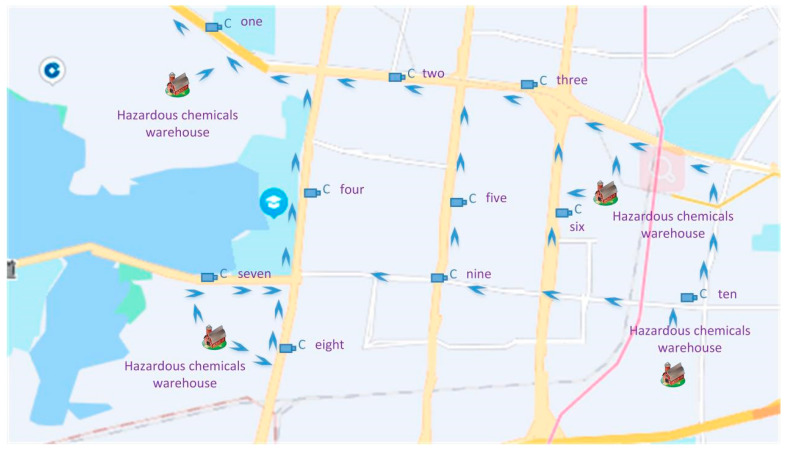
Four hazardous goods warehouses and ten CCD cameras on ten locations.

**Figure 10 sensors-22-07123-f010:**
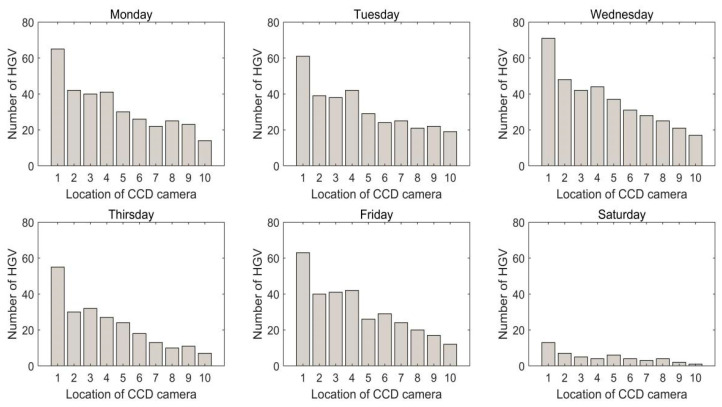
The number of hazardous goods vehicles of different position from Monday to Saturday.

**Figure 11 sensors-22-07123-f011:**
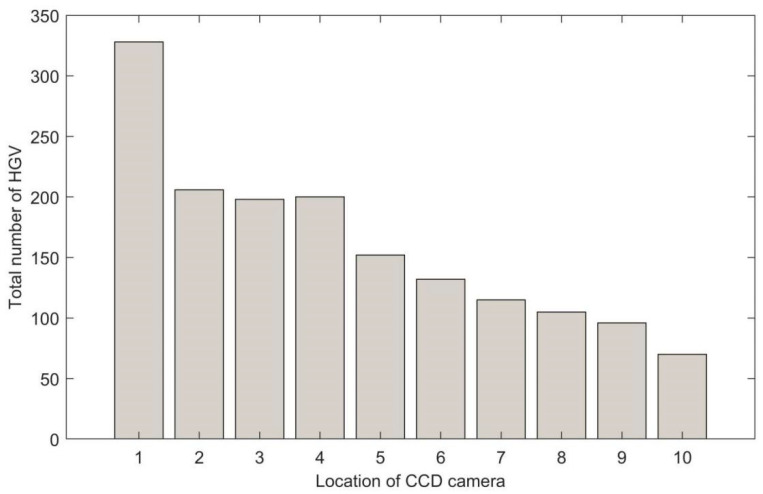
Total number of hazardous goods vehicles on each location from Monday to Saturday.

**Figure 12 sensors-22-07123-f012:**
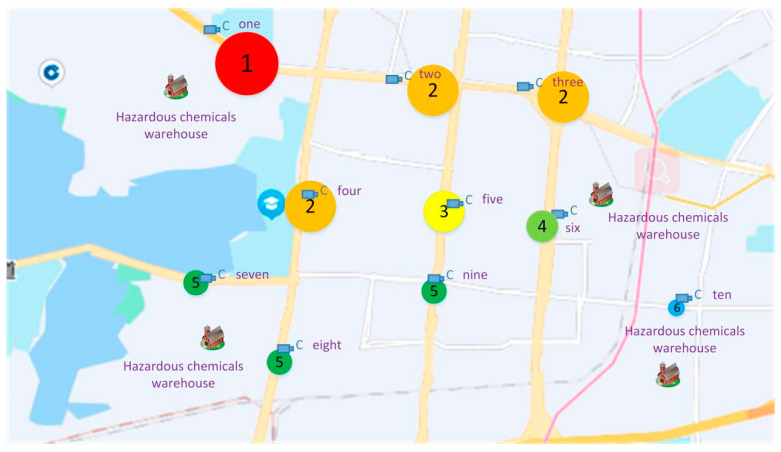
Different risk levels of different locations.

**Table 1 sensors-22-07123-t001:** Performance of different identification network models.

Different Recognition Network Models	Speed/ms	COCO mAP [^1]
CascadeR-CNN_ResNet-101	410	42.8
CenterNet_DLA-34	31	41.6
RetinaNet_ResNet-101	32	39.9
EfficientDet-D1	16	40.5
EfficientDet-D3	37	45.6
EfficientDet-D7x	285	55.1

**Table 2 sensors-22-07123-t002:** Four possible outcomes of the vehicle detection.

Positive (Presence of Fire)	Negative (Absence of Fire)
True Positive(TP)	True Negative (TN)
False Positive(FP)	False Negative(FN)

**Table 3 sensors-22-07123-t003:** Training time and accuracy of the two methods.

	EfficientDet-D3	Improved EfficientDet-D3
Training time (h)	6.3	4.2
Training accuracy	0.987	0.986

**Table 4 sensors-22-07123-t004:** Tri-partite measures of original and improved EfficientDet-D3.

Different Methods	Precision (%)	Recall (%)	F1-Score (%)
EfficientDet-D3	96.1	96.1	96.1
Improved EfficientDet-D3	97	97	97

**Table 5 sensors-22-07123-t005:** The number of parameters for the four different methods.

	Cascade R-CNN	CenterNet	EfficientDet-D7x	Proposed Method
Parameter (MB)	345	185	77	12

**Table 6 sensors-22-07123-t006:** The number of four possible outcomes.

Different Methods	TP	TN	FP	FN
Cascade R-CNN	95	38	6	7
CenterNet	96	37	7	6
EfficientDet-D7x	100	39	5	2
Proposed method	99	41	3	3

**Table 7 sensors-22-07123-t007:** Tri-partite measures of different methods.

Different Methods	Precision (%)	Recall (%)	F1-Score (%)
Cascade R-CNN	94	93.1	93.5
CenterNet	93.2	94.1	93.6
EfficientDet-D7x	95.2	98	96.6
Proposed method	97	97	97

## Data Availability

The data used to support the findings of this study are available from the corresponding author upon request.

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
