# Peer review of "Intelligent Detection of Hazardous Goods Vehicles and Determination of Risk Grade Based on Deep Learning"

_sensors, 2022, doi:10.3390/s22197123_

Round 1

Reviewer 1 Report

The paper is rather poor written, the significance of the result is not enough emphasized.

A revision of the equations should be perform to explain all used variables.

The expression in English should be carefully revised; few examples: page 2 - for the sake of the environment; it is difficult to en-65 sure that the vehicle is not blocked; the controllability problem is addressed for a two-time-scale discrete-time system; page 3 - According to∅=3; page 7 - Among them, the detection scores of the two hazardous 200 goods vehicles by this method are 100% (they are 98% and 99%)

In the abstract it is stated that "the proposed method is superior to the other two methods from three aspects of computational complexity, time consumption and detection accuracy", but in fact there are only two different aspect, while list computational complexity and time consumption means the same think (low complexity leads to low time consumption). This idea is also present in the Table 2 and Fig. 7.

Fig. 4 and Fig. 5 must be updated with variables on their axes. Fig 4 and Fig 5 should be arranged one under the other to easily see the correlation between them.

It is stated "The data set is divided into training data set and verification data set. Among them, 2387 images are training data sets and 146 images are verification 135 data sets." But no independent test set, that must not include any image from the training and verification sets is used for final performance analysis. An analysis using an independent test set must be included.

The detection score analyzed in Fig.6 for only two vehicle is irrelevant. Results should be presented for a much larger number of vehicle in an independent test set,  that does not include images used in the training process.

The analysis of the detection performance (Fig. 8), should be presented for a much larger number of vehicle in an independent test set,  that does not include images used in the training process.

In the Case study some conclusion are drawn regarding the number of vehicles passing to each check point (location). But this result is obvious from the route configuration and warehouses placement. No sophisticated method is necessary. Please explain the necessity to use the proposed method for that.

Round 2

Reviewer 1 Report

The paper can now be accepted for publication

Author Response

Thank you for your affirmation